# Computing Offloading Strategy in Mobile Edge Computing Environment: A Comparison between Adopted Frameworks, Challenges, and Future Directions

Shuchen Zhou [1,2,*], Waqas Jadoon [1,*] and Iftikhar Ahmed Khan [1]

1 Department of Computer Science, Comsats University Islamabad, Abbottabad Campus, Islamabad 22060, Pakistan; iftikharahmed@cuiatd.edu.pk
2 Institute of International Education, Huanghuai University, Zhumadian 463000, China
* Correspondence: 20101192@huanghuai.edu.cn (S.Z.); waqas_jadoon@cuiatd.edu.pk (W.J.); Tel.: +86-0396-2853018 (S.Z.)

**Abstract:** With the proliferation of the Internet of Things (IoT) and the development of wireless communication technologies such as 5G, new types of services are emerging and mobile data traffic is growing exponentially. The mobile computing model has shifted from traditional cloud computing to mobile edge computing (MEC) to ensure QoS. The main feature of MEC is to "sink" network resources to the edge of the network to meet the needs of delay-sensitive and computation-intensive services, and to provide users with better services. Computation offloading is one of the major research issues in MEC. In this paper, we summarize the state of the art in task offloading in MEC. First, we introduce the basic concepts and typical application scenarios of MEC, and then we formulate the task offloading problem. In this paper, we analyze and summarize the state of research in the industry in terms of key technologies, schemes, scenarios, and objectives. Finally, we provide an outlook on the challenges and future research directions of computational offloading techniques and indicate the suggested direction of follow-up research work.

**Keywords:** mobile edge computing; computation offloading; average delay; energy consumption

## 1. Introduction

In recent years, studies regarding the computational offloading of Mobile Edge Computing (MEC) have received a lot of attention from researchers [1]. From the emergence of MEC in 2014 to the end of December 2022, the number and proportion of papers related to computational task offloading have gradually increased, and the scope of the application of MEC has gradually expanded. The research resulted in increasing the attention and influence of computing task offloading.

The concept of computing task offloading in MEC was put forward by European Telecommunications Standards Association (ETSA) in 2014. Furthermore, the issues regarding computing task offloading in mobile systems—such as problems related to system economic cost, energy consumption, delay, or both energy consumption and delay—are outlined by Kumar et al. [2]. The MEC system has limitations in different application scenarios, such as limited computing or communication resources due to the limited computing task response time and complex application scenarios. Therefore, researchers have proposed computing task offloading methods for different MEC scenarios.

To ensure the quality of service (QoS), the mobile computing model has shifted from traditional cloud computing to MEC. The main feature of MEC is to "sink" network resources to the edge of the network to meet the needs of delay-sensitive and computation-intensive services and provide users with better services. Computation offloading is one of the major research issues in MEC. Although some scholars have conducted research on computational offloading schemes, there is a lack of systematic theoretical analysis from

the perspective of MEC. In this paper, the authors summarize the state of the art in task offloading in MEC. The main contributions of this paper include the following:

- Introduce the basic concepts and typical application scenarios of MEC and formulate the task offloading problem;
- Analyze and summarize the state of research in the industry in terms of key technologies, schemes, scenarios and objectives;
- Provide an outlook on the challenges and future research directions for computational offloading techniques and indicate directions for follow-up research work.

The rest of the paper is organized as follows: In Section 2 we introduce background and application scenarios of MEC; in Section 3 we provide a detailed discussion on the key technologies, schemes, scenario inter-task relationships and objectives; in Section 4 we debate the challenges; and finally, in Section 5 we discuss the future research directions and conclude the survey.

## 2. Overview of Mobile Edge Computing

### 2.1. Background of Mobile Edge Computing

ETSA proposed a MEC architecture in 2014 to provide mobile devices (MDs) with an IT environment and cloud-computing capabilities under the radio access network (RAN) [3,4]. ETSA has defined the MEC reference architecture in the literature [5]. As shown in Figure 1, the reference architecture includes two layers of MEC: the host level and the MEC system level [6]. Among them, the MEC host level is composed of a virtualized infrastructure manager, a MEC platform manager, and a MEC host. The MEC system level includes operator support systems, user application life-cycle management agents and MEC orchestrators. In addition, there is also a network level under the MEC system, which mainly includes local networks, external networks, third-generation cooperation plan (3GP) cellular networks and related external entities representing the access situation of the MEC system and the local area network, external access network or cellular mobile network [7]. Naouri, et al. [8] proposed a three-layer task offloading framework, which is composed of a device layer, cloudlet layer, and cloud layer, which can achieve high performance compared with advanced computing offloading technology.

MEC enables the traditional RAN to have business localization, low-latency, high-bandwidth transmission capabilities, and conditions for short-distance deployment services, effectively alleviating the pressure of mobile networks on transmission bandwidth and delay. Business downturn (that is localized deployment) can effectively reduce network load and bandwidth requirements, thereby saving network-operating costs and improving network resource utilization. The computation offloading flow diagram are shown in Figure 2.

### 2.2. Application Scenarios of Mobile Edge Computing

As one of the key techniques in MEC, computational offloading techniques have been applied in many scenarios [9]. Since MEC is close to the end devices and has certain computing power and storage capabilities, it has found applications in 5G, IoT, vehicular IoT, unmanned aerial vehicles, virtual reality/augmented reality, and other application domains where computation offloading is a key task [10,11].

#### 2.2.1. Internet of Things

The MEC can be used to process and aggregate small packets generated by IoT services before the data packets generated by the devices reach the core network. This will enhance the flexibility and scalability of IoT applications and connectivity, which is important for battery-powered IoT devices. The use of MEC reduces the transmission time between servers and devices, thus reducing battery consumption and improving the sustainability of devices and services, hence supporting long-term business.

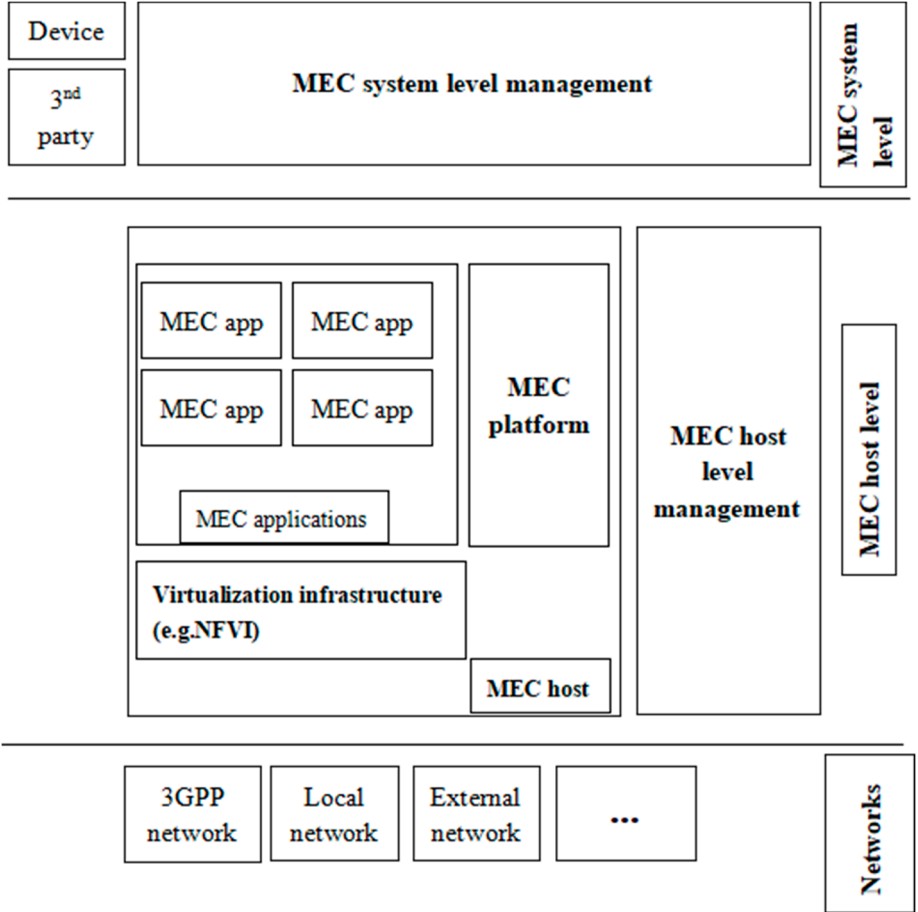

**Figure 1.** Multi-access edge computing framework.

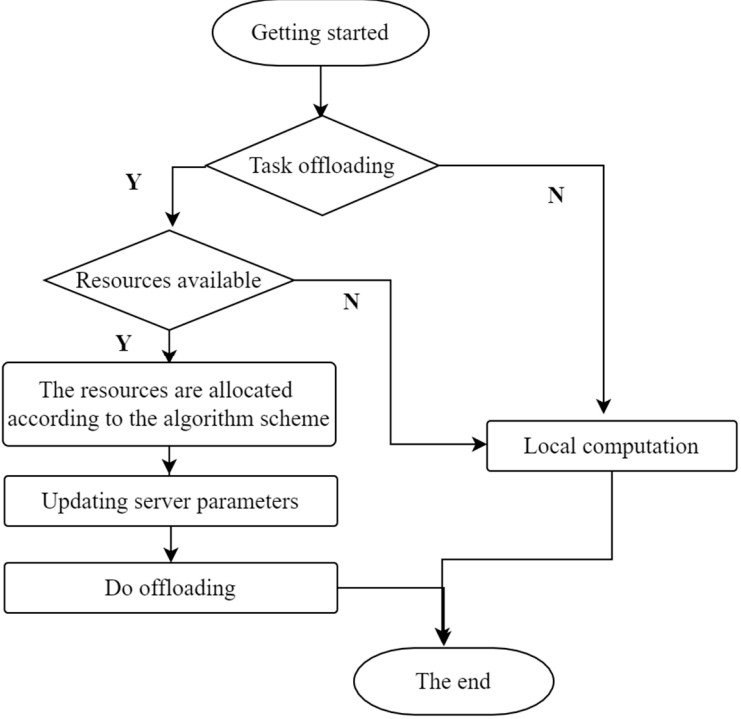

**Figure 2.** Computational offloading flow diagram.

IoT Analytics—a research organization focused on IoT, Machine to Machine (M2M) and Industry 4.0—released an IoT platform report in December 2019: The number of active IoT devices worldwide will reach 10 billion in 2020, and it is predicted that the number will reach 22 billion in 2025. These IoT devices collect a large amount of data, assuming all the data is uploaded to the cloud service center for processing, The remote cloud will experience significant pressure as a result. However, most IoT devices do not have the ability to process data or have poor computing power. The MEC can be used to process and aggregate small packets generated by IoT services before the data packets generated by the devices reach the core network [12]. This will enhance the flexibility and scalability of IoT applications and connectivity, which is important for battery-powered IoT devices. The use of MEC reduces the transmission time between servers and devices, thus reducing battery consumption and improving the sustainability of devices and services, hence supporting long-term business. For example, Fang, et al. [13] began their study using Mobile Edge Computing (MEC) of the IoT as their starting point, and they investigated the computation offloading strategy of UEs.

### 2.2.2. Smart Building Services

Nowadays, many smart buildings are evolving with their built-in technologies, and most of these smart building technologies require or generate local data. Therefore, to facilitate data communication between processing devices, a local IoT gateway is used. MEC is used for computing and local control data, including access control, climate and temperature control, smart signaling, asset tracking and security [14].

Edge computing moves computation and storage from core data centers to small micro-facilities closer to users, enabling faster processing and storage of data. This implies that sensors, devices and other gateways have the ability to operate locally instead of always relying on a core data center or cloud environment. The edge computing architecture is in line with the overall efficiency goal of smart buildings. Not only does it reduce latency, but since reaction times are faster and decisions can be made in real-time, there are significant cost savings because the data is also processed faster. To better understand edge computing and its role in the smart ecosystem, Deng et al. [15] analyzed the different use cases that make up the edge ecosystem to better understand these differences and their impact on the supporting infrastructure.

### 2.2.3. Smart Devices

Mobile users are increasingly demanding features and services supported by mobile smart devices. Sucipto et al. [16] proposed and implemented a computing offloading framework based on Near Field Communication (NFC) and designed a new NFC communication protocol that eliminates the need for constant user interaction which is a constraint and a limitation of one-way communication on data transmission. Another solution, such as that of Golkarifard et al. [17], is a new Device-to-Device (D2D)- based universal code offloading system for making offloading decisions. On the one hand, it can expand the capabilities of smart device services to meet the needs of users. On the other hand, offloading to edge servers (ES) instead of remote cloud servers can significantly reduce latency and device energy consumption and improve service quality and user satisfaction with the product experience.

### 2.2.4. Smart Grid

Smart grids require consumption and real-time monitoring of the important equipment involved in the entire process through sensors. Data is then collected through sensors for aggregation and analysis, and ultimately optimization of the power system [18]. Sensors can only perform some basic data filtering functions, while other tasks can be offloaded to ES for storage and preliminary aggregation, and finally offloaded to remote cloud-computing centers to complete tasks such as business intelligence data analysis [19].

In the above process, sensors can only complete some basic data filtering functions, and the remaining tasks can be offloaded to the edge server for storage and initial aggregation, and finally offloaded to the remote cloud-computing center for business intelligence data analysis and other tasks [20].

### 2.2.5. Smart Healthcare

By utilizing IoT in healthcare, we can enhance the lives of patients admitted to hospitals. Security and privacy play a major role when manufacturing devices, interconnecting things, and communicating while handling and storing data. Some of the primary functions of healthcare IoT include remote health monitoring, wearable devices such as sleep trackers, smart-shirt, smart-bracelet, and smart-watch on the body for monitoring and self-assistance, infusing medicine into the patient in a personalized way, and maintaining medical equipment [11].

Computational offloading techniques have also been applied in practical work in smart medicine and healthcare. For example, Chang et al. [21] designed a food recognition system for meal evaluation based on edge computing, which offloads the recognition task to nearby ES or cloud servers to solve the problems of delay and energy consumption. For example, real-time monitoring and feedback are required to avoid stroke patients' falls. This type of application data should be offloaded to ES for analysis, reducing latency and fast response to events. For example, Hennebelle et al. [22] proposed HealthEdge, a machine learning-based smart healthcare framework for type 2 diabetes prediction in an integrated IoT-edge-cloud computing system.

### 2.2.6. Internet of Vehicle (IoV)

Vehicle-to-infrastructure is a cutting-edge technology that may demand minimal waiting time and localized data processing. By connecting vehicles with networks according to communication protocols and standards to form the IoVs, the dynamic information of all vehicles can be collected, analyzed, and utilized to provide different services for running vehicles. IoV has stringent delay requirements, and excessive delays can lead to security, application and privacy data issues within the IoV ecosystem [21]. In the IoV, vehicles are required to maintain a constant connection with the server and engage in frequent data exchange. In the traditional cloud computing service, the cloud is far away from the vehicle and a large number of vehicles connect to the cloud as nodes, which will bring pressure on the cloud server for communication and database load. MEC can provide real-time and reliable vehicle connection, communication and security services. Through computation offloading technology, the service computation is offloaded to edge nodes, which can provide efficient and low-latency service quality.

MEC has proven to be one of the most effective technologies for delivering this capability. MEC technology can be further developed to provide extremely low latency, which is critical for 5G services [23]. For example, MEC services enable traffic control and smart parking; real-time warning of road conditions (congestion ahead, road bumps or icing); coordinate lane changes for vehicles, etc. [24,25].

### 2.2.7. Blockchain

Blockchain (BC) [26] is a distributed ledger (database) technology that connects data blocks in an orderly manner and cryptographically ensures that they are tamper-resistant and unforgettable. In general, BC technology can achieve openness, transparency, non-tamperability, unforgeability and traceability of all the data information in the system without third-party endorsement. As an underlying protocol or technical solution, BC can effectively address the trust problem and enable the free transmission of value. It has broad prospects in digital currency, financial asset transaction settlement, digital government affairs, certificate storage, and anti-counterfeiting data services. However, BC has huge scalability barriers that limit its ability to support frequent transaction services. However,

in the MEC scenario, scaling distributed cloud resources and services at the edge of the network presents significant challenges in terms of decentralized management and security.

Researchers have pointed out that the integration of BC and edge computing into a system can realize reliable access and control of storage and computation on the edge network, to provide large-scale network servers, data storage and validity calculation under security [27]. The application of MEC to BC technology enables the system to have many computing or storage resources distributed at the edge of the network, thus reducing the burden of BC storage and mining computation on power-limited devices. In addition, off-chain storage and off-chain computation at the edge can be implemented on the BC for scalable storage and computation.

### 2.2.8. UAV (Unmanned Aerial Vehicle)

The MEC environment provides services at the edge of the network. The UAV can first send data to the edge layer for data processing. Computational offloading techniques are used to offload data or service data collected by UAVs to ES, effectively reducing transmission and data processing delays.

UAV is an unmanned aircraft controlled with radio remote control equipment and programs [28]. Edge intelligent computing refers to the offloading of computationally intensive tasks generated by user nodes to edge servers with stronger computing capabilities for processing. Unmanned aerial vehicle (UAV)-based edge intelligent computing combines intelligent drone platforms on this basis and utilizes them with the advantages of strong mobility and easy deployment, Thus, it can provide edge computing services for ground user equipment more quickly and flexibly. At the same time, drones can also be used as user nodes to offload their computationally intensive tasks to the ground edge server for execution. For security issues, the positioning of UAVs by MEC edge nodes combined with cloud computing can be used to ensure the safety of UAV flights.

For instance, Wang et al. [29] introduced a trajectory control algorithm that utilizes deep reinforcement learning. The algorithm considers the inverse of the overall energy consumption of all users as a reward; and based on the current environmental state and real-time information, the UAV plans its next flight path to maximize the reward and minimize the total energy consumption of users. In addition, in the scenario of the UAV network, intelligent methods can be deployed on the UAV. In the study by Zhang et al. [30], the UAV employs a two-step process: First, it preprocesses the captured images using a Deep Learning model to extract relevant information. Subsequently, the UAV transmits this data to the ground-based edge server for further analysis. This approach effectively reduces the communication load to a significant extent.

### 2.2.9. Augmented Reality

Augmented reality (AR) is the overlay of a user's real-world view and the attached computer-generated image or any given input [31]. The input signals can be visual images, video, sound, graphics or even GPS data. Accurate values and readings from devices like cameras or position detectors are crucial for the proper functioning of AR services. Therefore, the low latency provided by MEC helps to meet these requirements. The advantage of using the MEC platform on the cloud is that it provides extremely localized data related to points of interest. To provide accurate information to MEC, AR requires ultra-low latency and a high data processing rate.

Tracking is a critical process and a widely studied research area in AR, as it enables a better understanding of the correlation between the user's (camera's) perspective and the surrounding environment. One of the tracking techniques used in AR is simultaneous localization and mapping (SLAM), which involves creating a map of the surrounding environment while navigating within it. SLAM enables the camera to autonomously construct a spatial map in real-time using acquired features and determine its position relative to the environment. Park et al. [32] proposed a new tracking system that integrates SLAM with a marker detection module for real-time AR applications in static and dynamic environments.

### 2.2.10. Social Networks

The proliferation of users on social and mobile platforms has led to significant advancements in the development of social networks. Social networks provide users with a platform to connect regardless of geographical or application barriers. Nowadays, with the development of next-generation mobile networks, social networks can provide users with various efficient applications, such as chat services, recommendation systems, decision support systems, game modules, etc., without affecting the connection between users [33].

Using the live broadcast scenario as an example, in the context of MEC and 5G, the video data is initially transmitted to edge servers (ES) located near the base station. These servers are responsible for synthesizing and processing the video data, creating versions with various resolutions, and delivering them to users based on their specific requirements. Users can choose the appropriate resolution channel to watch according to their needs, which meets the individualized needs and experience of the user.

### 2.2.11. Intelligent Video Acceleration

Intelligent video acceleration is commonly employed to enhance the quality of the end-user experience and leverage wireless network capabilities. Hypertext transmission protocol and transmission control protocol (TCP) are usually used for downloading, browsing and delivering Internet files or other media. The speed of TCP is not enough to adapt to the rapidly changing wireless access network conditions, which may lead to poor use of available radio resources, thus bringing a bad experience to users. In this case, the MEC server can provide the throughput required by the downlink and uplink interfaces and provide congestion control technology for TCP.

## 3. Research on Mobile Edge Computing Offloading

### 3.1. Computation Offloading Scheme

Computing offloading is a process of migrating computing tasks on MDs to the extended cloud or grid platforms, assisting terminal devices to complete user task requests, and returning computing results to designated devices [13]. In the MEC system, users use the MEC server as an intermediary to call the computing power closer to the local area for data processing, thus avoiding the traditional way of transmitting data to a remote central cloud for processing [34]. The flowchart of the MEC system computing offload is shown in Figure 3. A computing task can be executed locally or at the edge server. Local execution requires the mobile device itself to perform the computation of the task and output the result of the computation. The remote execution first needs to submit an offloading request, and the MEC server judges the validity of the user of the service: If the user is valid, the MEC server allocates computing resources to the user according to the resource usage and the user task data volume, and updates the computing resources and the system energy consumption status; if the user is illegal, the service cannot be provided. The server completes the tasks uploaded by the mobile device according to the allocated computing resources and returns the computing results to the device side.

Heuristic algorithms are widely employed to address the challenge of task scheduling in edge environments and to ensure the efficient allocation of resources for user-requested tasks. These algorithms provide practical and effective solutions by employing rules of thumb, guidelines, or approximations rather than exhaustive search methods. Basic prototypes include Grey Wolf Optimizer (GWO), Ant Colony Optimization (ACO), Particle Swarm Optimization (PSO), Genetic Algorithm (GA), etc. The comparison of computer offloading schemes is shown in Table 1.

While heuristic algorithms can discover viable solutions within constraints, they tend to have slow convergence speeds and may become trapped in locally optimal solutions during the solution process. As a result, it becomes challenging to fulfill the requirements of low-delay tasks due to the unpredictable deviation between feasible and optimal solutions. Hence, in this context, the reinforcement learning method [40] is leveraged to address the scheduling problem within a complex edge environment. By continuously correcting the

deviation between feasible solutions and optimal solutions, the decision-making capability of reinforcement learning not only accelerates the convergence speed but also enhances the overall solution quality. However, due to its own limitations, it cannot deal with high dimension and continuity problems. Deep Learning methods focus on the expression of perception and input and are good at discovering the characteristics of data. Because Deep Learning can make up for the shortcomings of Reinforcement Learning, Deep Reinforcement Learning (DRL) [43] uses the ability of deep neural networks to capture environmental characteristics and the decision-making ability of Federated Learning (FL) to solve complex system control problems [42]. And edge nodes can be used as intelligent agents to learn scheduling policies without global information about the environment.

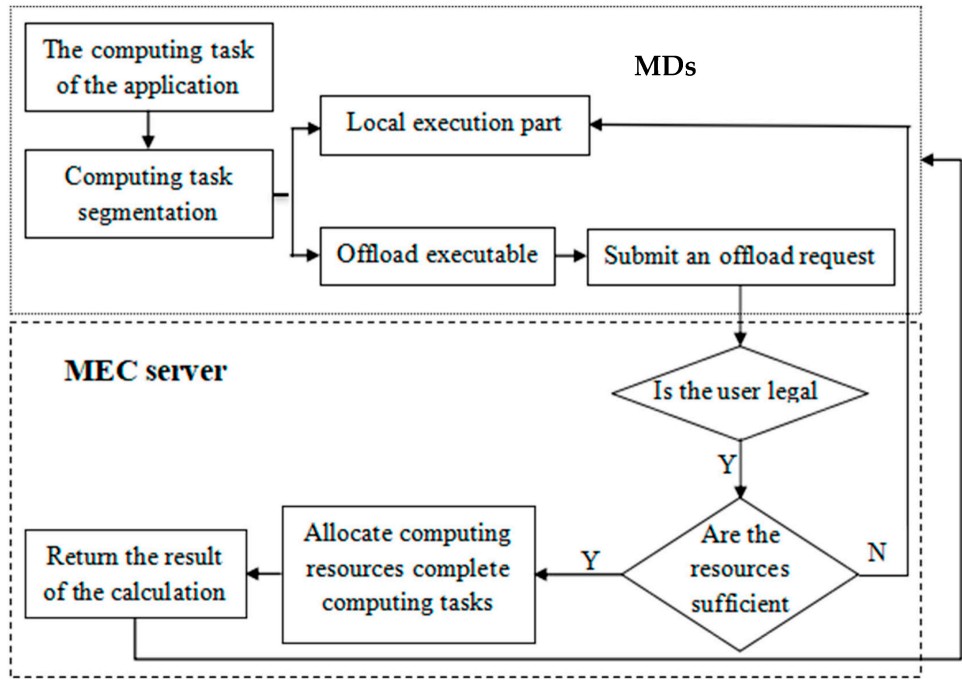

**Figure 3.** Task offloading process.

**Table 1.** Comparison of computer offloading schemes.

| Algorithm | Ref. | Granularities | Objective | Research Contents |
|---|---|---|---|---|
| GWO | [35] | Full offloading | Delay & energy | Used GWO's natural heuristic approach to achieve optimal solutions |
| ACO | [36] | Partial offloading | Energy | Proposed QoS based resource allocation and scheduling has used swarm-based ant colony optimization provide more predictable results |
| PSO | [37] | Partial offloading | Delay | Proposed multi-objective scheduling based on PSO under this research to give optimal allocation for a large number of tasks |
| GA | [38] | Partial offloading | Energy | Proposed an adaptive particle swarm optimization algorithm based on genetic algorithm operator |
| Q-Learning | [39] | Partial offloading | Delay | Proposed a deep Q-learning based autonomic management framework |
| DRL | [40] | Partial offloading | Delay | Proposed a new DRL-based online computing offloading scheme that considers both blockchain data mining tasks and data processing tasks |
| FL & DRL | [41] | Partial offloading | Delay & energy | Proposed FL-combined DRL to optimize the computational offloading scheme and edge content cache in MEC system and adopt DDQN method |
| FL | [42] | Partial offloading | Delay & energy | Proposed a FL-based approach to train preference models while retaining user data on their personal devices |

### 3.2. Granularities

Offloading granularities enable different levels of tasks to be offloaded in the cloud or on the edge server. These granularities can be classified into two categories, including fine grain and coarse grain. That is, full and partial offloading [9]. The process of offloading execution by granularities is detailed in Figure 4. A comparison between the literature on compute offloading classification in MEC systems is shown in Table 2.

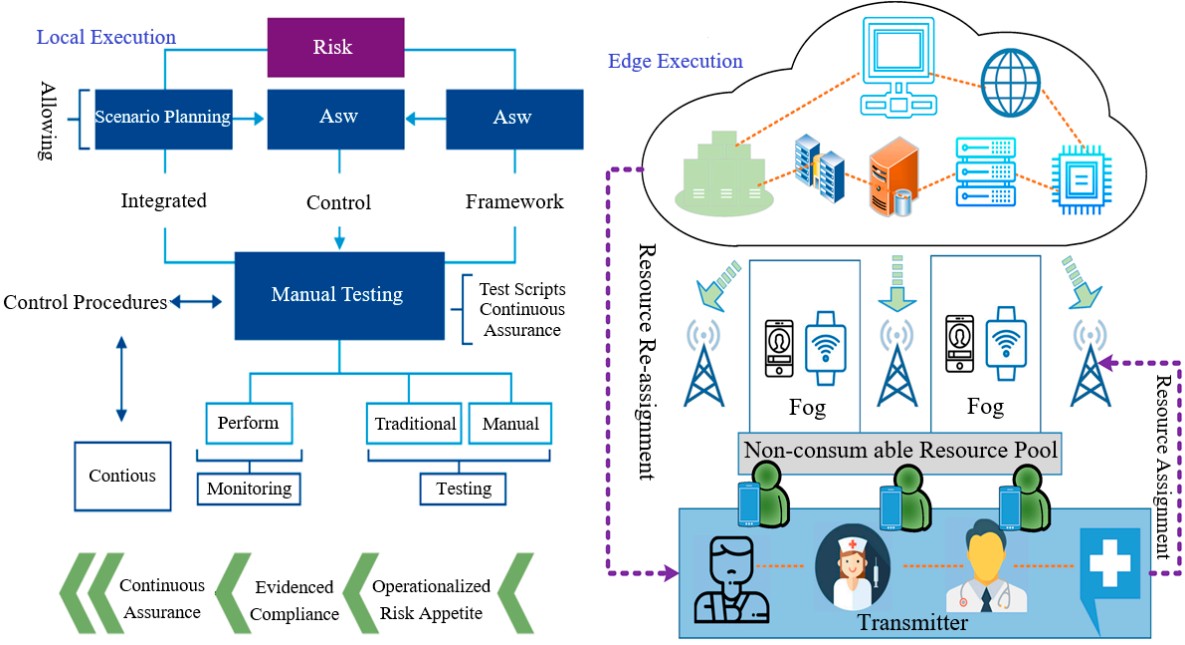

**Figure 4.** Full and partial offloading.

**Table 2.** Comparison between the literature on computing offloading types in MEC systems.

| Classification | Ref. | Platform | Objective | Research Contents |
|---|---|---|---|---|
| Full Offloading | [44] | MEC | Delay & energy | Provide Task offloading Service in Multi-Access Edge Computing Environment with Iterative Distributed Algorithms |
| | [45] | MEC | Delay | The application of several stochastic models of optimal stopping theory in offloading decisions is studied |
| | [46] | V2X MEC | Delay & energy | Study the joint computation offloading and URLLC resource allocation strategy for collaborative MEC assisted cellular-V2X networks. |
| | [47] | MEC | Delay & energy | Analytically addresses computation offloading strategy optimization with multiple heterogeneous servers in MEC. |
| Partial Offloading | [48] | MEC | Delay | we investigate the collaborative computation offloading, computation and communication resource allocation scheme |
| | [49] | fog computing | Delay & energy | Proposes a partial offloading method based on replicator dynamics of evolutionary game theory |
| | [50] | MECC | Delay & energy | propose an improved game-theory-based particle swarm optimization algorithm to obtain task offloading strategies |
| | [51] | MEC | Delay & energy | The joint problem of autonomous MEC servers' operation and MDs' QoS satisfaction in a fully distributed IoT network |

#### 3.2.1. Full Offloading

Full offloading [49,52]: The computation task is highly integrated or relatively simple, which cannot be partitioned. The whole computation task is offloaded and processed by the ES. The full offloading model aims to minimize the time overhead (latency), and energy overhead with time constraints, or to achieve a balanced trade-off between them. Giorgos et al. [44] adopted the data offloading framework based on the pricing mechanism,

constructed the prospect theory utility function, and proposed an iterative distribution algorithm for the unmanned aerial vehicle (UAV)-assisted multi-access edge computing systems. However, the data offloading decision-making problem is addressed by only considering the computational resources and ignoring the transmission characteristics, such as the delay rate and energy consumption of computing devices. Thus, depending on the environment, the wireless communication aspects between the UAV and users may have an impact on the offloading process. The principles of Optimal Stopping Theory (OST) are used by Iteration et al. [45] to address the task offloading decision-making problem for minimizing the expected processing time of task execution. The suggested OST model can be effectively applied in mobile nodes to make decisions according to the independent selection of mobile nodes and achieve the goal of minimizing the total delay when offloading tasks. The main goal of this work was to have a minimized total delay when offloading a task without considering the energy consumption constraints. Lei et al. [46] proposed a joint computing offloading and ultra-reliable and low-latency communication (URLLC) resource allocation strategy for the cooperative MEC-assisted cellular Vehicle to Everything (v2x) network. Considering the significance of both reliability and delay in vehicle communication, the suggested method formulates a joint power consumption optimization problem while preserving network stability, thus achieving the goal of energy consumption. Li [47] tried to achieve the balance among the minimization of time overhead, minimization of energy overhead, and the minimization of cost-performance ratio for the single user in multiple MEC environments. Meanwhile, more studies were conducted to minimize the energy overhead while satisfying the latency constraints. Labidi et al. [53] devised two solutions for the computation offloading decision process, aiming to minimize the average energy consumption of single MD, and guaranteeing the average delay service of the applications on the MD. Furthermore, Sardellitti et al. [54] minimized the overall mobile device user's energy consumption while meeting latency constraints by applying the convex approximation technique. The proposed method jointly optimized the wireless and computational resources for a multi-wireless scenario. The disadvantage is that it only focuses on static frameworks and a single cloud. Some academics focus on the strategies for achieving an appropriate trade-off between time overhead and energy overhead. Chen et al. [55] adopted game theory to formulate the computation offloading decision process where the utility function was the weighted sum of time overhead and energy overhead. The balance between time overhead and energy overhead was obtained by minimizing the utility function. However, how to jointly formulate the power control and offloading decision-making process remains unaddressed.

3.2.2. Partial Offloading

Partial offloading [49,52]: The computation task can be divided into several subsets. Then, a part of the computation task is processed locally while the rest is offloaded to ES for processing. The literature on partial offloading of the computation tasks mainly divided the objectives into two categories: minimization of energy overhead with latency constraint, and a balanced trade-off between time and energy overhead. Kai et al. [48] have examined both the offloading of collaborative computation and resource allocation for computation and communication schemes in a multi-MEC scenario. The authors developed a framework for collaborative computing that allows MDs' tasks to be partially processed at cloud center (CC), edge nodes (EN) and terminals while considering computation and communication capacities, respectively. Further, a pipeline-based offloading scheme is introduced to offload tasks from EN with limited computing resources to the CC with sufficient computation resources. A non-convex optimization problem is formulated to minimize the sum latency of all the MDs for the developed approach, and a successive convex approximation (SCA) approach is used to address the formulated problem. However, due to the finite computation and communication capacities at EN, the power consumption at network and task latency may increase as the number of users increases. A partial offloading strategy based on replicator dynamics of evolutionary game theory to jointly optimize latency and energy

consumption has been proposed in the literature [49]. A disadvantage of this scheme is that it is difficult to achieve a stable equilibrium point with an increasing number of users due to the large solution space. Wang et al. [50] proposed an improved game-theory-based particle swarm optimization algorithm to obtain task offloading strategies. The proposed scheme can reduce the processing time and save system energy consumption. However, the security of the mobile user needs to be strengthened. Apostolopoulos et al. [51] formulated the joint problem of autonomous MEC servers' operation and MDs' Qos satisfaction as a minority game. Tran et al. [56] adopted the mixed integer nonlinear program to model the joint problem of task offloading and resource allocation for maximizing the users' task offloading gains. Zhao et al. [57] presented the computation offloading algorithm for the multi mobile device users (MDUs) scenario using nonlinear programming with high time complexity. Moreover, Zhao assumed the MDUs had the same channel quality and the same computing capabilities. This case is not realistic for the real network. You and Huang [58] designed the partial computation offloading process based on the different channel qualities, the energy overhead of MDs, and the fairness among the MDs. Meanwhile, they gave a sub-optimal offloading decision algorithm, but this solution would lead to negligibly higher energy overhead of MDs compared with the optimal solution. The abovementioned paper on partial computation task offloading minimized the MDs' energy overhead depending on the quality of the WC and the transmission power of the MDs.

### 3.3. Key Technologies of MEC

To implement MEC and make it usable, several key techniques are described below [59]:

### 3.3.1. Software-Defined Networking (SDN)

The core idea of introducing SDN is to allow the use of products and off-the-shelf hardware to create programmable and application-aware intelligent networks [60], which is achieved by separating the control plane that manages the network from the data plane that transmits the actual data streams. A well-defined open interface between the two planes is key to ensuring interoperability between various device manufacturers and suppliers. The centralized SDN controller helps to solve some classic network problems, such as routing, tunneling and IP address translation, as well as new challenges in future 5G applications, such as user mobility, adaptability to service degradation, application-specific security and the integrated protection of the IoT systems [61]. Through SDN, network traffic can be flexibly controlled to seamlessly integrate MEC computation and caching into providing network services for mobile applications [62].

### 3.3.2. Network Function Virtualization (NFV)

NFV leverages virtualization technology to achieve flexible network function design, deployment and management, regardless of the underlying physical network devices [63]. These network functions may include classic functions (e.g., firewalls, deep packet inspection), elements of the Evolved Packet Core (EPC) (i.e., a framework for providing converged voice and data over LTE networks), as well as innovative functions, such as network coding, data aggregation or compute-as-a-service. The intuitive extension of the NFV concept combines the functions of a single virtual network to realize the modularization of complex functions in the service function chain (SFC) [64].

### 3.3.3. Information-Centric Networking (ICN)

The Internet, originally used for host-to-host communication, is now used primarily for content distribution. Information-centric networks are designed to bridge the gap between the original design of the Internet and current emerging applications—such as HD video-on-demand streaming, 3D gaming and augmented and virtual reality—with increasing traffic. To optimize caching and content distribution, ICN recommends that the Internet architecture be redesigned as a content-centric network, which uses two design

concepts, namely, networking named content and intra-network caching on MEC servers, to reduce bandwidth pressure and improve data transmission [65].

*3.4. Scenario of Computing Offloading*

With the rapid development of wireless networks, their topologies are becoming more and more complex, which complicates the research scenario of computing task offloading methods in MEC systems. According to different system scenarios, this paper divides the existing research on computing offload into three scenarios: single-user single-server, multi-user single-server and multi-user multi-server, and classifies the existing representative research work based on this standard to obtain the results shown in Table 3 for different computing offload scenarios in MEC system. As can be seen in Table 3, in the research work of MEC computing task offloading, the focus and solution of computing task offloading decisions vary by the scenario. Therefore, to fully understand the content and core of the research work on computational offloading in different scenarios, this section analyzes and summarizes the main research work in the table.

**Table 3.** Comparison between the representative literature based on different scenarios for computation offloading in MEC.

| Classification | Ref. | Objective | Research Contents |
|---|---|---|---|
| Single user single server | [66] | Energy | This paper proposes a method of joint allocation of CPU and network resources and tasks |
| | [67] | Delay & energy | Joint optimization of task offloading scheduling and transmission power allocation in MEC system |
| | [68] | Delay & energy | Determine the best computation mode according to the energy consumption when tasks are executed locally and offloaded |
| Multi user single server | [69] | Energy | Minimize device energy consumption under the constraint of task cache stability through tradeoff between device energy consumption and task delay in multi-user MEC system |
| | [70] | Delay & energy | Under wireless conditions, online task scheduling and model optimization can be performed simultaneously to minimize delay and energy consumption. |
| | [71] | Delay & energy | Joint optimization of multi-user offloading decisions and allocation of shared communication resources |
| Multi user multi server | [72] | Energy | Research on computing task segmentation and collaborative offloading of intelligent IoT applications |
| | [73] | Delay & energy | This paper studied the task decomposition collaborative offloading for load balancing MEC |
| | [74] | Delay | This paper proposes a cross edge computing offloading framework for sharable applications |

3.4.1. Single User Single Server

For the research on computing task offload of single-user single-server MEC system [75], most works only determine the corresponding task offload decision based on the offload target. For example, DREAM [66] studied the balance between computing energy consumption and processing delay in MEC systems under non-offload load, cloud offload load, network traffic and other heterogeneous application environments. Based on Lyapunov optimization, a new algorithm was proposed to jointly determine the offload strategy, task allocation on local and remote processors, CPU clock speed and interface selection between Wi-Fi and cellular networks. The AT-based method [67] mainly solved the problem of offload scheduling and joint allocation of transmission power of multiple independent computing tasks in the MEC system. The algorithm proposed in the literature, based on flow shop scheduling and the convex optimization method, can minimize the weighted sum of task execution delay and device energy consumption, which further proves that the most significant delay performance improvement can be achieved when the effective radio and computing resources are relatively balanced. The SWIPT-based method [68] optimized the time interval required for energy acquisition and information

decoding during local computing, as well as the time slot and power required for computing task offloading and determined the optimal processing mode of computing tasks based on the lowest energy consumption required for computing task processing on low-power devices. The research of MEC task offloading in the single-user and single-server scenario is relatively simple, and there are usually few factors that affect the offloading decision. Generally, only one indicator of local execution and edge server processing needs to be judged.

### 3.4.2. Multi-User Single Server

For the research on computing task offloading of multi-user single-server MEC systems, since computing tasks on multiple MDs need to compete for limited wireless communication resources and edge server computing resources, such research usually needs to consider the coordination and scheduling among multiple MDs. Specifically, Mao et al. [69] constructed the problem of minimizing device energy consumption under the constraint of task cache stability based on the random arrival of computing tasks on multiple MDs. Online algorithms based on Lyapunov optimization design in the literature can obtain the optimal CPU cycle frequency during local execution and the optimal bandwidth allocation and transmission power during offload execution by balancing the energy consumption and task delay of MDs. Considering that multiple MDs may produce severe signal interference when offloading computing tasks simultaneously through a small cellular network, which may affect data transmission during the offloading task, HuZ et al. [70] adopted the federated learning technology to minimize the weighted sum of the energy and time consumption in High-altitude balloons with MEC enabled. Chen W et al. [71] aim to save the energy consumption of MDs while ensuring the QoS of all users. The algorithm not only considers the offload decision of computing tasks on multiple users but also considers the competition for shared communication resources when multiple users offload tasks at the same time. The research contents of the above representative literature show that the offloading of computing tasks in a multi-user MEC system not only focuses on the allocation of computing resources of the mobile device itself, but also considers the competition of multiple users for shared resources such as communication, server cache and computing, which also provide an effective solution for dealing with the resource coupling relationship between multiple devices.

In addition, in the actual industrial production environment, cloud enterprises expect that a single edge server can serve as many industrial MDs as possible to ensure the user's QoS and data security while reducing system costs and investment in the operation process.

### 3.4.3. Multi-User Multi-Server

For the research of MEC system computing offload in multi-user and multi-server scenarios, not only the competition of limited communication and computing resources among multiple MDs should be considered, but also the cooperative services among multiple ES, that is the way in which multiple ES provide services for multiple computing tasks on MDs. For example, in the application segmentation and collaborative computing offloading of the IoT network, Chen M-H et al. [72] solved the problem of minimizing the energy consumption of all MDs, considering the common interests of task requesters and collaborators in terms of energy consumption, while making the goal meet the delay constraints of a single task. In the study of the Genetic Algorithm (GA) method [73], the problem of cooperative computation offloading among multiple MEC servers in 5G HetNets (Heterogeneous Networks) was considered. This practical problem was first constructed as a constrained optimization problem with the goal of reducing the energy consumption and processing delay of MDs. At the same time, the GTOS algorithm was proposed to solve the problem based on game theory. Chen W et al. [74] proposed a cross-ES computing offloading framework for divisible applications which considers the collaboration between multiple ES when MDs move and which is based on factors such as task transmission cost, processing cost, collaboration cost between servers and punishment in case of task

failure. At the same time, the online Lyapunov optimization algorithm provides a feasible solution for the energy acquisition of MDs and the selection of ES for computing tasks in computing offloading. Based on the analysis of the representative literature, in the MEC system composed of multi-user and multi-server, the decision of computing offloading should consider not only the mobility of MDs and the collaborative processing of the same computing task among multiple ES, but also the benefits of collaboration between MDs themselves and ES. Therefore, there are still many challenging problems to be solved for MEC task offloading in this scenario. In addition, with the rapid development of 5G communication technology, ultra-dense IoT networks will become the deployment trend in the industrial Internet environment.

### 3.5. Comparison of Inter-Task Relationship of MEC Task Offloading

In the study of computing task offloading of MEC system, there are three relationships among multiple computing tasks of data shareable or code shareable applications, namely, sequential dependent tasks, parallel dependent tasks, and general dependent tasks. Their respective topological relationships are shown in Figure 5 [76].

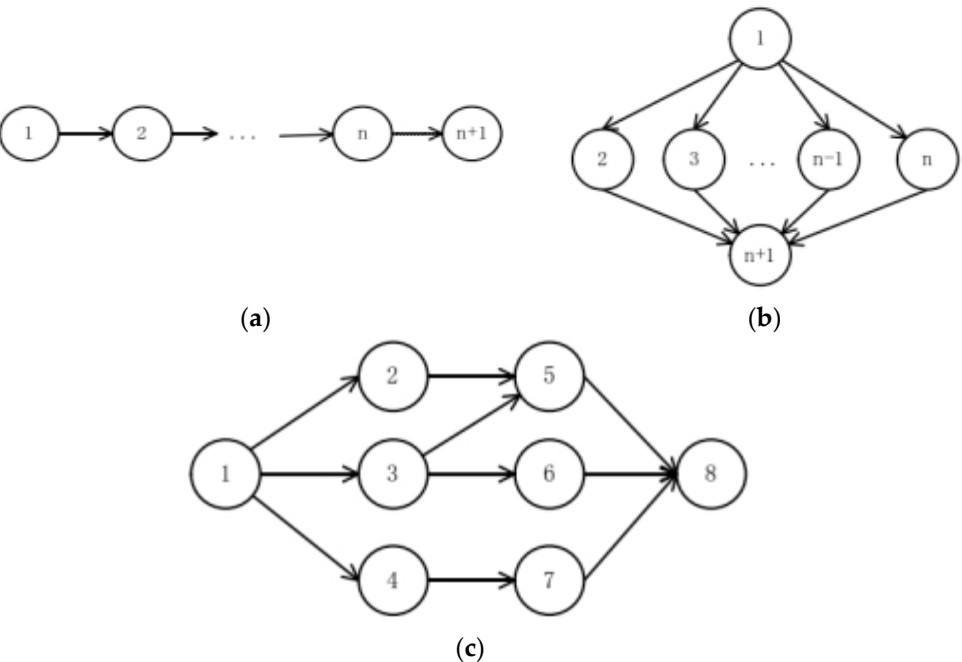

(**a**)     (**b**)

(**c**)

**Figure 5.** The topology graph of relationships between offloading tasks in MEC, should be listed as: (**a**) Sequential dependent; (**b**) Parallel dependent; (**c**) Generally dependent.

Specifically, sequence-dependent computing tasks can only be scheduled and processed sequentially. This is because each of the subsequent computing tasks has a predecessor and a successor task, except for the start task and the exit task. The scheduling execution of the successor task usually depends on the computing results of the predecessor task. Therefore, in the multiple computing tasks that depend on the sequence, if you want to reduce the completion time of the entire application, you can only offload all the computing tasks in the entire application to an edge server that has more storage capacity and computing power than the mobile device itself. For parallel dependent computing tasks, other computing tasks are independent of each other except the start task and the exit task. In this case, you can consider offloading some computing tasks to the edge server for processing, while the other part can be processed in parallel on the local processor. Finally, for computing tasks with general dependencies, there are not only sequential dependencies but also parallel dependencies between computing tasks. There are many factors affecting the offloading scheme of computing tasks, which makes it more challenging to determine the offloading decision of computing tasks.

Based on the above dependencies among computing tasks, it can be seen that the offloading decision of computing tasks with dependencies in the MEC system is usually affected by different task scheduling orders. There are usually the following relationships among tasks with dependencies: (1) the computation result of the predecessor task is the input data of the successor task; (2) due to the limitation of software and hardware, some computing tasks can only be executed locally, such as the start task, exit task, data display task of the application, etc.; (3) different scheduling orders of parallel computing tasks will have different effects on the overall completion time of the application. Therefore, when offloading dependent computing tasks in the MEC system, it is necessary to consider not only the general factors related to offloading but also the scheduling sequence of each computing task. For example, Mahmood et al. [77] first combined the scheduling of computing tasks with offloading and proposed a JSCO algorithm to determine the optimal offloading decision of each computing task. The solution to the problem has more degrees of freedom by using the task scheduling order of wireless sensing to replace the scheduling order predefined by the compiler. Meanwhile, because the application program on the mobile device is composed of multiple program modules, the algorithm can reduce the completion time of the application by deploying appropriate modules on the mobile device and executing them in parallel in the cloud. Liu et al. [78] studied the scheduling and offloading problem of computing tasks on the IoVs, in which the computation-intensive applications on the vehicle can be divided into multiple interdependent sub-tasks and can be offloaded to the edge server of the roadside unit for processing. To complete the application within a specified time and minimize the average completion time of multiple applications, the MAMTS algorithm was proposed to solve the task scheduling problem on multiple MEC servers. Sundar et al. [79] studied the scheduling decision problem of computing tasks in a general-purpose cloud computing system composed of local processors and remote cloud servers. To minimize the execution cost of an application composed of multiple interdependent computing tasks within the deadline, the ITAGS algorithm was proposed in the literature. The algorithm obtains the optimal scheduling and offloading decision through the binary relaxation of the original problem and the greedy optimization of each task in the allowed time. A comparison of these existing works is shown in Table 4.

**Table 4.** Comparison between computation offloading tasks dependencies in MEC.

| Existing Algorithm | Research Contents | Restrictive Factors |
|---|---|---|
| JSCO [77] | Joint optimization of scheduling and offloading of applications tasks | Equipment consumables, communication delay, overall execution time and task dependency |
| MAMTS [78] | This paper studied the scheduling problem of multiple computing tasks on multiple ES in the IoV | Task dependency, completion time, and average completion time are minimized |
| ITAGS [79] | Determine the scheduling decisions of interdependent computing tasks such that the completion time of the whole application is minimized under the constraint of the deadline | Task dependency and application deadline |

Computing tasks with dependencies exist in different application scenarios. With the rapid development of wireless networks, the network topology of MEC becomes more complex, which makes the research on computing task offloading with known dependencies popular. In addition, when multiple interdependent computing tasks are offloaded, different scheduling orders, judgment indexes and execution modes will have a greater impact on the offloading decision of the tasks. Therefore, it is of practical value and significance to study the scheduling and offloading of multiple computing tasks with dependencies in a MEC system composed of multi-user and multi-ES.

### 3.6. Objectives of Computing Offloading

By analyzing a large number of references of existing MEC task offloading schemes, we can find that most of the references use standard metrics that affect the overall efficiency of the system, among which the more important indicators include energy consumption, delay, response time and system operating cost [80]. In the existing literature, these indicators are either considered separately or jointly as a common multi-objective problem. Research work spans mainly three different optimization goals: (1) minimizing latency; (2) minimum energy consumption; (3) tradeoff between delay time and energy consumption. In this paper, we divide the computational offloading schemes into the three categories mentioned above, compared the computational offloading schemes with the same objective and analyzed the similarities and differences, advantages and disadvantages of the proposed schemes. Related work sets different optimization objectives for different MEC application scenario requirements and offloading methods to obtain the optimal computational offloading scheme for a particular scenario. Table 5 shows some research progress in recent years.

**Table 5.** Summary of research on task offloading.

| Objective | Ref. | Platform | Scheme | Scenario | The Solution and Contribution |
|---|---|---|---|---|---|
| Delay | [81] | MEC | F | MUMS | Proposes an online offloading and resource allocation algorithm based on Lyapunov optimization theory |
| | [82] | MEC | P | MUMS | Proposed an algorithm that can reduce latency and improve task fairness |
| | [83] | MEC | F | MUMS | Proposed A offloading decision generation algorithm based on deep reinforcement learning |
| | [84] | MEC | F | MUSS | Proposed solution algorithms under different bandwidth constraints |
| | [85] | MEC | P | MUMS | The computation overhead model is built based on game theory |
| Energy | [86] | MEC | P | MUMS | The machining path is adopted, and the energy consumption is minimized by the continuous convex approximation method. |
| | [38] | MEC | P | SUSS | Proposed an adaptive particle swarm optimization algorithm based on genetic algorithm operator |
| | [87] | Fog & MEC | P | SUSS | Designed a heuristic algorithm to find the optimal offloading scheme under delay constraints |
| | [88] | MEC | P | MUSS | Proposed a task offloading algorithm based on convex optimization theory and Gibbs sampling |
| | [89] | MEC | P | MUSS | A dynamic offloading and resource scheduling optimization model for a multi-user mobile edge cloud SWIPT system |
| Delay & energy | [61] | Fog & MEC | P | MUMS | Proposed a joint computation offloading and radio resource allocation algorithm |
| | [90] | MEC | P | MUMS | Transforming the problem into a Markov decision process and designing algorithms based on deep reinforcement learning |
| | [91] | MEC | P | MUMS | Proposed a multi-objective evolutionary algorithm to solve the optimal offloading scheme to balance energy consumption and delay |
| | [92] | MEC | P | MUMS | Proposed an improved algorithm based on MOEA/D |
| | [93] | MEC | P | MUMS | Adopt the distributed idea from game theory to study the computation offloading problem |
| Profit maximization | [94] | MCC | P | MUMS | Proposed a new hierarchical model to provide computing resources based on auction profit maximization |
| | [95] | MEC&CC | P | MUMS | Nash equilibrium is achieved based on game theory to maximize the utility of CS and ES |
| | [96] | MEC | P | MUMS | Propose a breakeven-based double auction and a more efficient dynamic pricing based double auction |
| | [97] | MEC | P | MUMS | Propose a truthful combinatorial auction mechanism |
| | [98] | MEC | P | MUMS | Propose an incentive mechanism in a non-competitive environment |

F—Full Offloading, P—Partial Offloading, MUMS—Multiuser multi-server, MUSS—Multiuser single server, SUSS—Single user single server.

### 3.6.1. Minimize the Average Delay

In the evaluation criteria of computing task offloading, the processing latency is usually defined as the time required for the computing task to be executed locally, the transmission time required for the computing task to be transmitted to the edge server, the processing time required by the computing task at the edge server and the time required to return the computing result to the mobile device [99]. According to the definition of the processing delay, the factors such as the size of the calculation task requested on the mobile device, the transmission bandwidth of the wireless channel, the channel state of the wireless link, the distance between the mobile device and the edge server, the CPU cycle frequency of the edge server and the mobile device, and the number of CPU cycles required by the size of each computing task unit are all factors to be considered in the calculation of this index, and the determination method of each factor is the same as that of each factor in the energy consumption index. This optimization objective is crucial for the performance guarantee of delay-sensitive mobile applications. Aiming at long-term network energy efficiency, literature [81] proposed an online offloading and resource allocation algorithm based on Lyapunov optimization theory, which can provide better efficiency and energy-delay in multi-user and multi-server task offloading. The advantage of this algorithm is that while optimizing the delay, the energy consumption of the base station is very low through the power limitation and base station sleep strategy.

Literature [100] based on the smart grid scenario, given the limited computing resources power terminal high time delay problem, when processing applications, to offload tasks and resources allocation are studied, constructs the mixed integer nonlinear programming (MINLP) problem. Firstly, the Lagrange multiplier method is used to get the optimal solution of resource allocation. Under this premise, the improved self-adaptive genetic algorithm (AGA) is used to get the effective task offloading scheme. Simulation results show that the proposed algorithm can significantly reduce the total terminal delay compared to other baseline algorithms. Similarly, the MINLP problem is also established in the literature [82]. The difference is that the tasks in this paper can be computed not only on local or ES but also on other terminals. After simulation verification, the proposed CTS (Comprehensive Task Scheduling) algorithm can improve the fairness between the task and reduce the total delay, but it is a shortcoming of high computational complexity. In the literature [101], Markov's theory was used to analyze the average delay of tasks and the average power consumption of devices, and the problem of delay minimization under power constraints was proposed. The optimal solution is found based on a one-dimensional search algorithm, which effectively solves the problem. The literature [83] constructed WPMECN (Wireless Powered MEC Network) model; in this model, based on the depth of reinforcement learning is proposed an uninstall decision generation algorithm, compared with the baseline algorithm. This algorithm has a significant advantage in CPU processing delay but has little difference from the baseline algorithm in task average delay. For delay-sensitive applications in the IoT, the literature [84] proposes an approximate algorithm and an online algorithm without bandwidth constraints, as well as a heuristic algorithm under bandwidth constraints, to obtain the offloading decision under the minimum delay. Simulations demonstrate that the proposed algorithm has promising applications.

To minimize the overall data computation and transmission time at all levels, the literature [85] focuses on the partial computation offloading problem for multi-user in an MEC environment with a multi-wireless channel. The computation overhead model is built based on game theory. The advantages of this algorithm are that the delay is low and the processing rate is high, but the disadvantage is that the computation offloading of any proportion adopted has strong assumptions. The literature [102] studied a multi-user secure MEC system, in which users computing offload tasks to the base station in the presence of eavesdroppers. To minimize the total computing and transmission delay of all the users under the constraints of security and computing resources, this paper jointly optimized the transmission power, computing resource allocation and user connection mode of MDs, and designed a block coordinate descent, continuous convex approximation

algorithm and branch-cut methods. The advantage of this algorithm is that it has low time complexity, which reduces the delay while taking into account the system security and finds a basic trade-off between the delay and security. The disadvantage is that the allocation of wireless spectrum resources is ignored. The literature [103] studied cellular network assisted MEC systems. Traditional cellular households allow MEC users to use their licensed frequency channels to simultaneously offload computational load to MEC servers through non-orthogonal multiple access technology. In this scenario, the power and energy storage of cellular users and MEC users and the computing capacity of MEC servers are limited.

### 3.6.2. Minimize Energy Consumption

The energy consumption refers to the energy consumption of the mobile device itself when the computing task is executed locally, the energy consumption of the mobile device when the computing task is transmitted from the mobile device to the edge server, the energy consumption at the edge server when the computation task is processed at the edge server, the energy consumption of the edge server when the calculation result of the edge server is returned to the mobile device and the energy consumption of the mobile device itself when the calculation result is received by the mobile device [104,105].

This optimization goal alleviates the pressure of energy-intensive mobile applications on the limited battery capacity of MDs. Literature [86,106] proposed a MEC computing offloading scenario based on UAV communication. The MEC server was deployed on the UAV communication base station to provide a computing offloading environment for mobile users in the coverage area. The machining path was adopted, and the energy consumption was minimized by the continuous convex approximation method. The advantage of this scheme is that it can improve the QoS in harsh communication environments, but the disadvantage is that the research on the allocation of wireless spectrum resources and computing resources is relatively lacking. Literature [38] focuses on DNN (Deep Neural Networks) based intelligent IoT, based on self-adaptive particle swarm optimization algorithm musing the genetic algorithm operators (SPSO-GA), which is used to solve the offloading strategy to minimize the energy consumption under the delay constraint. Simulation results show that the policy obtained by the proposed algorithm performs better in terms of energy saving compared to the other baseline algorithms. Literature [87] in order to further reduce energy consumption of equipment, in addition to uninstall decision-making, considered the power control and computing resource allocation problems, modeling to turn it into a joint optimization problem, puts forward the E2PC (Energy Efficient Power Control) algorithm to find the optimal solution under delay constraint. Simulation results show the superiority of the algorithm in reducing energy consumption, but the disadvantage is that the algorithm complexity is too high. In reference [88], the task offloading decision with statistical QoS guarantee was studied, that is, the task completion time was allowed to exceed a given threshold within a certain range, and the energy consumption was reduced by reducing the QoS requirements. In this paper, we propose a task offloading algorithm using convex optimization theory and Gibbs sampling methods.

In the literature [89], a dynamic offloading and resource scheduling optimization model for a multi-user mobile edge cloud SWIPT system is developed through system computation and energy model analysis, and then this non-convex optimization problem is transformed into a zero pairwise gap optimization problem. With the digitized distributed computing migration algorithm, the results of this paper make it possible to minimize the system energy consumption while satisfying the optimal policy requirements for clock frequency control, transmission power allocation, offloading ratio and received power splitting ratio.

### 3.6.3. Optimize Energy Consumption and Delay

In the study of computing task offloading in the MEC system, different researchers use different methods and indicators to calculate the offloading cost, which can be time cost,

energy cost, or economic cost. Generally, cost refers to the cost of local MDs processing computing tasks, the cost of sending computing tasks by transmitting multimedia, the cost of processing computing tasks on the server side and the cost of receiving correct response results, etc. These costs usually depend on the configuration, response time and requirements of the task [107]. In this regard, since both latency and energy consumption are key factors in calculating the total cost, it is important to find a compromise between these two useful metrics.

Literature [61,108] independently optimized energy consumption and delay, and considered the energy consumption of delay-sensitive MA and MDs in MEC offloading. In [61], Chang et al. proposed a joint computation offloading and radio resource allocation algorithm based on Lyapunov optimization to minimize the system costs related to the latency, energy consumption and weights of MDs. In the literature [108] Chen et al. proposed to leverage artificial intelligence for next-generation wireless networks to effectively provide ultra-reliable low latency communications and pervasive connectivity for the IoT. Literature [109] introduced multi-access edge computing to solve the problem of choosing the optimal offloading decision for MEC servers distributed in ultra-dense networks. Online UE-BS and BS-learning algorithms were proposed to minimize average energy consumption while considering the cost. The disadvantage of this literature is that it does not consider the allocation study of computing resources. Literature [70] jointly optimized the energy consumption and delay to promote the research progress of energy consumption and delay trade-off optimization in MEC computing offloading. In the literature [70], to shorten the response time and reduce the energy consumption, an offloading architecture–Ternary Decision Maker (RDM), was designed. The advantage of this paper is that a detailed offloading architecture is designed by constructing an actual simulation environment, which makes a good contribution to computational offloading. The disadvantage is that its modeling method is complex and needs to be further optimized.

Literature [90] was designed based on the edge of 5G vehicle sense multiple access network (VAMECN), puts forward the system time delay and energy consumption optimization problem of minimizing the weighted sum, The mixed integer nonlinear programming problem into a Markov decision process, and then based on the depth of the reinforcement learning design Joint Computation Offloading and Task Migration Optimization, (JCOTM), simulation results show that the algorithm can effectively reduce the delay and energy consumption in different system environments. Literature [91] proposes a multi-objective evolutionary algorithm, whose core idea is to use evolutionary principles such as crossover, mutation and selection to find the Pareto frontier, which is the optimal offloading scheme that balances energy consumption and latency. Simulations have shown that this scheme is effective in reducing latency and energy consumption, and the more iterations, the better the results. Similarly, the literature [92] improved the evolutionary algorithm based on multi-objective decomposition for multi-objective optimization problems of joint task offloading, power allocation, and resource allocation, and proposed MOEAD_MEC algorithm, simulation results show that this algorithm is significantly superior to other baseline algorithms in reducing latency and energy consumption.

To optimize the compromise between energy efficiency and delay, a Stackelberg game model is established in the literature [93] to jointly optimize the proportion of load and bandwidth allocation. The advantage of this algorithm is that it has good performance in dynamic environments, and ultimately achieves the goal of weighting and minimizing energy and time costs. The disadvantage is that the research work on the selection of MDs needs to be strengthened.

In summary, current research on task offloading in edge computing usually aim to minimize a certain index (delay, energy consumption or the weighted sum of both), establish mathematical models about the delay and energy consumption, consider relevant constraints, design algorithms and solve the task offloading strategy. However, due to the different scenario requirements, data heterogeneity and network instability, to find the optimal solution, it is necessary to consider concurrent issues such as resource allocation

and power control while solving the task offloading strategy. In terms of algorithm design, most of the existing literature uses mathematical optimization algorithms such as convex optimization and Lyapunov optimization, as well as heuristics such as genetic algorithms and ant colony algorithms to solve task offloading strategies.

### 3.6.4. Economic Benefit

The abovementioned literature review mainly optimizes the weighting of delay, energy consumption, delay and energy consumption without considering the economic benefits. In studies considering the social benefits of MECs, most work mainly uses auction algorithms or other incentive mechanisms to achieve optimal benefits. The proposed approach [94] introduces a novel hierarchical structure that enables a bidirectional bidding and asking interaction between the mobile device requesting computing offloading and the edge server. This structure facilitates the auctioning of computing resources and communication resources, aiming to maximize profits while ensuring the Quality of Service (QoS) standards for users are met. The literature [95] establishes a fresh cooperative relationship between the CC server and the MEC server, leveraging game theory principles as the foundation for this collaboration. Task offloading is carried out from the top layer to the bottom layer, and the task offloading is awarded by bidding. By verifying the Nash equilibrium relationship between them, the existence of the optimal task offloading quantity and bid is proved, and the optimal solution is found through several iterations. The same relationship applies to edge servers and mobile devices. The literature [96] studied how to enhance social benefits in the industrial Internet of Things, modeled the two-way interaction between MEC servers and mobile devices, and constructed a general dual auction framework to solve interaction problems and maximize social benefits. On the premise of satisfying various economic attributes, the real dual auction algorithm is used to sell the computing resources of the MEC server to optimize the benefits. In the literature [97], the competitive transaction between the MEC server and the user is modeled using a three-tier auction framework. The entire auction process is divided into three separate and sequential sub-problems. To address this, a novel real combinatorial auction algorithm is proposed, which aims to optimize both the service cost and wireless channel resources simultaneously. In the literature [98], an incentive mechanism is established by optimizing market pricing profit, without taking into account channel resources. Additionally, an online multi-round auction algorithm is designed specifically to accommodate user randomness. In the literature [110], a many-to-many computing resource allocation algorithm based on double auction is proposed with the objective of maximizing benefits. The author examines the cross-server resource allocation scheme in MEC from a network economics perspective and models the bilateral interaction between the MEC server and user as a double auction. This approach addresses the challenge of cross-server interaction involving multiple user tasks.

### 3.7. Safety

MEC exhibits specific characteristics such as multisource heterogeneity, cross-trust domain and limited terminal resources. As a result, new security and privacy concerns may emerge, rendering traditional data security and privacy protection mechanisms used in cloud computing environments no longer applicable. The research on MEC security in IoT is summarized in Table 6. In the literature [111], a privacy-preserving aggregation scheme for MEC-assisted IoT applications is proposed. The scheme model encompasses terminal devices, MEC servers and public cloud centers. The process begins with the terminal device encrypting the collected data using a key, generating the corresponding signature and transmitting the ciphertext and signature to the MEC server. The MEC server, in turn, employs the terminal device's public key to validate the messages, aggregate the ciphertexts, generate the corresponding signatures and submit them to the public cloud server. Finally, the public cloud server verifies the validity of the MEC's ciphertext and decrypts it using the key. The proposed scheme ensures user privacy, offers source authentication and integrity and achieves nearly 50% communication cost savings compared to traditional methods.

**Table 6.** Summary of research on MEC security.

| Optimization Objective | Literature | Key Research Points |
| --- | --- | --- |
| Establish a trust evaluation mechanism | [111–114] | Propose a privacy-preserving aggregation scheme for MEC-assisted IOT applications<br>Propose a fine-grained trust evaluation mechanism for service selection<br>Dl-based multi-user physical layer authentication scheme<br>Gradient descent is used to accelerate the training of deep neural networks<br>Large-scale training of machine learning using data augmentation |
| Defending against attacks | [115–117] | proposes a K-neighbor joint optimization of task offloading<br>SPEA2 (improving the strength Pareto evolutionary algorithm) is employed to acquire balanced task offloading solutions<br>we propose an integer linear programming (ILP) model and a dynamic programming algorithm<br>we propose an integer linear programming (ILP) model and a dynamic programming algorithm |

In order to establish and maintain a unified and trusted IoT environment, literature [112] proposed a fine-grained trust evaluation mechanism for service selection based on edge computing. The edge computing platform filters service requests through a trust evaluation mechanism to resist malicious attacks. As a trusted third party, the platform is an effective service access point, which provides a trust evaluation and service selection mechanism for service selection and has a large number of storage and computing resources. The trust value records of device-related parameters are updated in real-time, and when an anomaly occurs, the edge platform can explain the reason according to the records. Experimental results show that the selection mechanism based on record criterion has the advantages of global, high stability, strong fault tolerance and easy management. Literature [113] proposes a multi-user physical layer authentication scheme based on DL, which uses machine learning methods to improve the physical layer authentication mechanism, greatly improving the security of the MEC system in IoT. Considering that the training of machine learning requires a large number of training samples, the training process is time-consuming and computing resource intensive. The gradient descent method is used to accelerate the training of deep neural networks and reduce computational overhead and energy consumption. Literature [114] proposed a multi-user physical layer authentication scheme with data augmentation, which re-generates the data set from the existing data set through certain combination operations and realizes the establishment of an accurate authentication model by using machine learning algorithms. Simulation results show that the proposed method improves the certification rate, speeds up training proficiency and reduces the training cost.

In the context of the Internet of Vehicles, reference [115] introduces an optimized privacy protection mechanism based on differential privacy technology. This mechanism aims to safeguard the context-sensitive information of terminal vehicles during the task offloading process. Additionally, it mitigates the impact of noisy data on offloading decisions, enhancing the effectiveness of time delay optimization. Considering the security and privacy issues caused by the accumulation of massive data at the edge nodes, literature [116] proposed a privacy entropy model to deal with the privacy protection problem through the method of quantitative privacy. Then, the load balance, transmission time and privacy entropy were comprehensively analyzed and modeled, and a multi-objective optimization problem was defined together. Finally, the problem was solved by a heuristic algorithm to obtain a privacy-preserving task offloading scheme. In addition, [117] proposed a task segmentation method to solve the privacy problem in the process of task offloading. The specific content includes that the task is classified according to different security requirements, and then the whole task is divided into multiple parts (high-security requirements tasks must be divided, and low- and medium-requirement tasks should be divided according to the number of system resources). Finally, each task block was

offloaded to multiple edge nodes under the same or different service providers according to the security requirements of the task itself.

## 4. The Challenges of Computation Offloading Technology

MEC has attracted a lot of attention due to its ability to significantly reduce the energy consumption of mobile terminal devices, which can implement real-time application offloading. However, computational offloading techniques in MEC are still not mature. This section is devoted to the analysis of the challenges faced by the computational offloading techniques. This section will present the challenges faced by the computational offloading techniques in terms of mobility of terminal devices, ES, security, user privacy data and service heterogeneity.

### 4.1. Mobility of Terminal Devices

In previous studies, the mobility of a terminal device can be divided into three categories: one is the assumed mobility path of the terminal device; the second is to predict the movement path of the terminal device via Markov decisions. The third is to predict the movement path via online learning. The movement of the terminal equipment is divided into two types: within and across the fixed MEC service area.

(1) Movement of terminal equipment within a fixed MEC service area. Within a fixed MEC service area, the energy required to transmit data changes significantly during offloading due to the degradation of channel quality caused by device movement. When computing tasks are not installed on an edge server, the device consumes more energy during installation and the service latency is higher.

(2) Movement of Terminal Devices across MEC server regions. One of the challenges of computational offloading technology is how to ensure service continuity when a mobile device being offloaded is moving from one MEC server region to another. In this case, if the movement path and route of the mobile device can be accurately predicted in advance, the transfer of offloading data is not carried out in a certain period across the MEC service area, but only the necessary computing service is performed locally on the terminal device or the computing service is suspended. Since the cross-MEC service area is the case where the mobile terminal devices are present during the mobility process, but the time required to cross the service area is smaller, the above scheme can theoretically solve the existing problem and scholars can do further research and discussion.

Another solution is for the server to forward the installation task data to the next MEC server before crossing the servant region after the predicted path is obtained. For example, Ouyang et al. [118] proposed a novel computational offload scheme for adaptive user management, which can optimize the perceived delay of users and the cost of computational offload, weighted by user preferences.

However, there is a certain amount of error in predicting the moving path. To predict the path more accurately, machine learning and other related artificial intelligence techniques should be used. However, AI-based solutions introduce a new set of issues such as training time and cost. it is challenging to trade off the optimal cost in computing offload effectiveness and path prediction.

### 4.2. Edge Server Mobility

When the terminal device is no longer fixed, but has a certain mobile path, in addition to considering the problem of the cross-MEC service area, we also need to consider the mobility of the edge layer server in the MEC. When the edge server moves, it is necessary to consider how to keep the continuity of service and meet the QoS requirements of users. The question is: how to efficiently handle the mobility path of the ES, offload the terminal computing task to the most suitable edge server, and satisfy the QoS. The movement of the edge server imposes a high load on the backhaul and leads to high latency, which makes it unsuitable for real-time applications. The mobility of ES is challenging due

to the limited communication between ES. At present, most studies on computational offloading schemes [84,85,89,93] only consider the energy consumption of terminal devices. However, the energy consumption of MEG servers should be further considered during the selection of offloading schemes. Another major challenge is how to account for backhaul between ES in the MEC and the ability to reflect their load and parameter changes during offloading. Scholars can make full use of the resource characteristics of MEC servers for further research, to improve the QoE of terminal equipment and optimize the performance of the whole system for mobile users.

### 4.3. Malicious Data on Terminal Devices

In IoT application scenarios, terminal devices collect a large amount of data, which may contain dirty data collected by malicious nodes. This dirty data may attack the MEC server or the cloud server, causing the server to crash. By deploying relevant models on the MEC server, the MEC server cleans the data transmitted from the terminal, retains sensitive data and screens malicious data [119]. Wang et al. [120] proposed a data cleaning model based on MEC, established a cleaning Neem based on SVM and optimized the model using an online search method. In the single-shot scenario of traditional IoT, the data feature dimension of the terminal device is relatively simple, and the computing capacity of the terminal device can meet the demand. The data dimension of the terminal device becomes too complex to support data filtering on its own. The above problem can be solved efficiently by offloading the data filtering task to the MEC server. Since MEC servers have a certain amount of storage space, it is feasible to deploy data-clearing models.

### 4.4. Safety

Security has always been an important topic in MEC, as well as in computational offloading techniques. Security is one of the areas worth studying. Computational offloading techniques in MEC face many security issues, such as edge server security, data security, network security, etc.

(1)  Edge server security. If a server providing services at the edge layer is attacked by a malicious server, it will make an error while providing services. Currently, most computing offloading schemes assume that the ES participating in the service is secure and reliable, which is an ideal state. In the following research work, we need to consider practical applications and further explore the security of ES. The current trust evaluation mechanism of ES can also be used to ensure the security of the server [121].
(2)  Data security. In the traditional cloud computing model, user data is completely stored in the cloud server, which is exposed to the risk of privacy disclosure [122,123], as well as tampering during data offloading to the peripheral layer, as well as privacy issues [124,125]. The data can be encrypted during transmission. The advanced encryption standard used in the literature [124] is the Reed Solomon coding process. In addition, access to data from terminal devices is also a part of data security. Since MDs are distributed in the MEC, if these devices are stolen, it should be considered that the relevant data will not leak private data even if obtained by a criminal.
(3)  Network security. Security issues arising from network factors, such as firewall systems and intrusion detection systems, need to be considered in the overall system for computing offloading. To protect the entire MEC computing offload system from attacks. Stolen private data.

In traditional cloud computing, solutions to security problems are gradually developed, while MEC, as an extension of cloud computing, allows scholars to learn from effective and mature solutions to relevant network security problems in traditional cloud computing. However, since MECs are distributed and the network environment is complex, the relevant solutions are not fully applicable. It is a big challenge to design an effective solution that can address the security issues in MEC based on the characteristics of computing offloading technologies.

*4.5. Isomerism*

The mobile network environment is complex and diverse. In the 5G era, the IoT is growing rapidly. With the emergence of a variety of new applications, the heterogeneity of networks and devices is an important issue. As science and technology evolve, there may be issues such as incompatibility of devices and different data types in offloading technical solutions. Currently, all existing computational offloading schemes assume the same MEC server for experiments and analyses, but this does not reflect the heterogeneity of real networks. When the designed offloading solution is not deployable on some emerging applications, the feasibility of computing the offloading solution is not high. Therefore, it is important to consider the heterogeneity of networks and devices in real-world scenarios when designing offloading schemes. Designing offloading solutions that allow access to different types of devices on different networks and maintain a high degree of compatibility is a major challenge.

## 5. Summary and Outlook

With the network changes brought by 5G, MEC has attracted a lot of attention as an architectural revolution. MEC can effectively solve various problems faced by clients and servers. While the study of MEC is socially motivated and valuable, MEC itself is still an immature and unproven technology. Computational offloading is one of the key technologies in MEC, which addresses the under-resourcing and energy consumption of MDs. Computational offloading schemes play a crucial role as it determines at which layer the task is executed, whether it is a local terminal device, an edge device, or both. At the same time, during the survey, it was also found that most of the current research is based on the ideal case of fixed MDs. Whether the investigated algorithms are still efficient requires further research and experimental verification. In different application scenarios, the latency and energy consumption requirements are different, and not one algorithm is suitable for all scenarios, so the performance and portability of the algorithms need to be discussed separately based on the actual scenarios.

This paper has reviewed the existing research on MEC system including computing offloading, resource allocation, etc., but there are still many aspects that needs further research and breakthroughs. The following is a summary of the follow-up research directions.

(1) With the popularity of the IoT, the number of devices that need to perform computing offloading has increased dramatically, and the network needs to have more capacity to accommodate more users. Due to the need for wireless data transmission during the offloading process, the shortage of wireless spectrum resources will make the number of users that can be accommodated in the network very limited. The performance gain obtained only by optimizing the allocation of wireless resources is bound to be limited. The next step is to combine non-orthogonal multiple access NOMA with MEC to accommodate more users, expand system capacity and improve system performance. On the other hand, it plans to open newer spectrums, such as millimeter wave, etc., which will be used for MEC to further improve the performance of users during computing offloading.

(2) The QoS constraints in this paper are all deterministic constraints, which require the task processing delay to be less than a given threshold. However, there are many applications in practice, such as mobile video, etc., whose QoS requirements can be given in the form of probability. For example, the probability that the delay is greater than a given threshold value is less than a certain value, that is, the soft delay requirement. The next step is to take soft delay constraints into account, making the problem model suitable for more practical applications. Although this paper has certain innovations in problem modeling and solution, these studies are still limited to the theoretical level. Following the principle of "practice is the only criterion for testing truth", in future work, we will focus on how to apply research results to actual networks, transform knowledge into productivity and make our own contributions to the advancement of MEC.

(3)  In future work, the joint optimization problem of communication and computing resources will be considered. In addition, artificial intelligence will be employed to optimize the computational offloading problem. Machine learning algorithms and power control techniques will be considered to solve the problem of computing offload in the real-time changing MEC environment.

**Author Contributions:** S.Z. and W.J.; methodology, W.J.; validation, I.A.K.; formal analysis, I.A.K.; investigation, S.Z. and W.J.; resources, S.Z.; data curation, I.A.K.; writing—original draft preparation, S.Z.; writing—review and editing, W.J.; visualization, I.A.K.; supervision, W.J.; project administration, W.J.; funding acquisition, S.Z. All authors have read and agreed to the published version of the manuscript.

**Funding:** Grant number "2021XJGLX67". This work was supported in part by the research grants provided by Huanghuai University.

**Data Availability Statement:** Data sharing is not applicable to this article as no datasets were generated or analyzed during the current study.

**Acknowledgments:** All authors would like to thank the anonymous referees for their valuable comments and suggestions.

**Conflicts of Interest:** The authors declare no conflict of interest.

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
