# Peer review of "Computing Offloading Strategy in Mobile Edge Computing Environment: A Comparison between Adopted Frameworks, Challenges, and Future Directions"

_electronics, doi:10.3390/electronics12112452_

Round 1
Reviewer 1 Report
The paper is devoted to the review of offloading strategies in mobile edge computing (MEC) technology. The authors reviewed 93 papers that have been appeared during the last decade, which are focused on researches of different aspects of MEC.
Generally, in my opinion, the paper (review) is well structured reflecting the goal of the review, i.e. offloading strategies in MEC. Nevertheless I have following comments:
1. The most strategies are taken into account. I only see just one aspect that should be analyzed and supplemented: there are plenty of publications dealing with security problems in MEC and different MEC offloading strategies have been proposed so far that reflect these very important problems. The authors spread sketchily security issues in various parts of the paper (mainly in challenges, section 4.4.). My suggestion is to add a separate subsection in section 3 dealing the offloading strategies in terms of security.
2. We can find similar reviews or surveys dealing with MEC problems. It would be valuable to reference to them with some comments (this is just one example: Computation Offloading in Mobile Cloud Computing and Mobile Edge Computing: Survey, Taxonomy, and Open Issues, https://doi.org/10.1155/2022/1121822)
3. I suggest to add references to example applications of MEC in section 2.2 – only some of them are referenced.
4. Line 805: “This paper has done some research on the MEC system including computing offload-805 ing, resource allocation, etc.,…” – The paper is just the review of researches.
5. And some editorial comments:
- Figure 2 is illegible – cannot be reviewed
- Figure 3 – Yes/No should be added near the rhombi
- Figure 4 is not required - all elements of partial offloading can be found in Figure 3. There is no sens to repeat them here.
- there are many typos in the paper – need to be corrected.
Moderate editing of English language is needed.
Reviewer 2 Report
1. What is the main question addressed by the research?
To ensure the QoS, the mobile computing model has shifted from traditional cloud computing to mobile edge computing (MEC). The main feature of MEC is to "sink" network resources to the edge of the network to meet the needs of delay-sensitive and computation-intensive services, so as to provide users with better services. Computation offloading is one of the major research issues in MEC. In this paper, the authors summarize the state of the art in task offloading in MEC.
2. Do you consider the topic original or relevant in the field? Does it address a specific gap in the field?
Yes. The problem solved in this paper is relevant to this journal.
3. What does it add to the subject area compared with other published material?
The main contributions of this paper include:
- Introduce the basic concepts and typical application scenarios of MEC and formulate the task offloading problem
- Analyze and summarize the state of research in the industry in terms of key technologies, schemes, scenarios and objectives
- Provide an outlook on the challenges and future research directions of computational offloading techniques and indicate directions for follow-up research work
4. What specific improvements should the authors consider regarding the methodology? What further controls should be considered?
- In section 2.2.7, how to apply blockchain technologies like blockshare a blockchain empowered system for privacy-preserving verifiable data sharing, vchain optimizing verifiable blockchain boolean range queries, bdns a secure and efficient DNS based on the blockchain technology, in different applications?
- In section 2.2.9, how to apply the existing method epar an efficient and privacy-aware augmented reality framework for indoor location-based services to improve the security of augmented reality?
- For SDN, offloading and MEC, more up-to-date works could be studied, such as new mobility-aware application offloading design with low delay and energy efficiency, dynamic trajectory planning for unmanned aerial vehicle based on sparse A* search and improved artificial potential field, real-time cache-aided route planning based on mobile edge computing, smartphone-assisted smooth live video broadcast on wearable cameras.
5. Are the conclusions consistent with the evidence and arguments presented and do they address the main question posed?
Yes.
6. Are the references appropriate?
More technical papers could be investigated.
7. Please include any additional comments on the tables and figures.
The tables and figures are clear and easy to understand.
The presentation of this paper is good.
Reviewer 3 Report
This paper reviews the state of the art in task offloading in MEC.
The authors introduced the basic concepts and typical application scenarios of MEC, and then they formulate the task offloading problem.
They have analyzed and summarized the state of research in the industry in terms of key technologies, schemes,
scenarios and objectives. Finally, they provided an outlook on the challenges and future research directions of computational offloading techniques and indicates directions for future research work.
The paper must undergo a major revision round.
-subsections 2.2.1 to 2.2.11 are very short, these subsections must be extended to explain how MEC can benifit these techonolgies.
-the text in figure 2 is not readable at all, quality of figure 3 and figure 4 must be improved as well.
- ' At present, most studies on computational unloading schemes only consider the energy consumption of terminal devices. when you make such statement, give some references to back it up
- a review of most used algorithms (grey wolf optimization, ant optimization ...etc) in the context of MEC computational offloading must be added.
- many recent related works are missing, for instance:
[1] Wang, Kun, Xiaofeng Wang, and Xuan Liu. "A high reliable computing offloading strategy using deep reinforcement learning for iovs in edge computing." Journal of Grid Computing 19 (2021): 1-15.
[2] Naouri, Abdenacer, et al. "A novel framework for mobile-edge computing by optimizing task offloading." IEEE Internet of Things Journal 8.16 (2021): 13065-13076.
and many other related works
Additional comments:
1. What is the main question addressed by the research?
This is a review paper, so there is no research question
2. Do you consider the topic original or relevant in the field? Does it
address a specific gap in the field?
No, the work is not an original research paper, it is just a review of the existing MEC task offloading methods
3. What does it add to the subject area compared with other published
material?
There is no significant added value to the subject area, as there are other papers that already reviewed task-offloading methods for MEC
4. What specific improvements should the authors consider regarding the
methodology? What further controls should be considered?
Add the paper selection methodology, how the papers were searched for, and what is the selection criteria
5. Are the conclusions consistent with the evidence and arguments presented
and do they address the main question posed?
Yes, to some extent
6. Are the references appropriate?
Many relevant references are missing
7. Please include any additional comments on the tables and figures.
The figures are of very bad quality, the text in figure 2 is not readable at all, quality of figure 3 and figure 4 must be improved as well.
Reviewer 4 Report
This paper presents a survey of computation offloading for mobile edge computing.
1. The literature about different applications of MEC-based computation offloading can be improved. Although the authors list different types of applications, more related studies should be included in Section 2.
2. It could be too late that the authors introduce the idea of computation offloading in Section 3.2. Since the paper discusses the stategies of MEC-based computation offloading, the key part should be described earlier.
3. The economic models of MEC-based computation offloading should be considered. Recently, several related literature have been published. These proposals design offloading strategies based on cost. I suggest the authors to include these research.
Reviewer 5 Report
Review comments regarding assigned manuscript entitled “Computing offloading strategy in mobile edge computing environment: A comparison between adopted frameworks, challenges, and future directions” given below:
1. The motivation is not clear in the Introduction section. The authors should analyze the limitations of existing studies related to computational offloading schemes and discuss the differences between the proposed work with these studies. Besides, the authors are suggested to summary the contributions of this study.
2. A Prisma flow diagram is required to depict the flow of information for meta-analysis.
3. The authors are suggested to make a punctilious taxonomy of mobile edge computing offloading in Section 3 if possible.
4. More diagrams are required for each point in Section 3.
5. The English editing should be improved as there are many typos and grammar mistakes.
Extensive editing of English language required.
Round 2
Reviewer 1 Report
The authors have improved the manuscript according to given suggestions. In my opinion, the revised paper raises all the major issues in the computing offloading in MEC.
Quality of English language in this paper is acceptable.
Reviewer 3 Report
my comments were reasonably addressed, i have no further comments
Reviewer 4 Report
The revised manuscript has addressed the concerns raised by the reviewer. I suggest to accept this paper based on its current form.
Reviewer 5 Report
The authors have incorporated all my concerns. There are no further requirments from my side.
The English editing has been improved significantly.